# AutoAssist: A Framework to Accelerate Training of Deep Neural Networks

**Jiong Zhang** [*]
zhangjiong724@utexas.edu

**Hsiang-Fu Yu** [†]
rofu.yu@gmail.com

**Inderjit S. Dhillon**[*†]
inderjit@cs.utexas.edu

## Abstract

Deep Neural Networks (DNNs) have yielded superior performance in many contemporary applications. However, the gradient computation in a deep model with millions of instances leads to a lengthy training process even with modern GPU/TPU hardware acceleration. In this paper, we propose AutoAssist, a simple framework to accelerate training of a deep neural network. Typically, as the training procedure evolves, the amount of improvement by a stochastic gradient update varies dynamically with the choice of instances in the mini-batch. In AutoAssist, we utilize this fact and design an *instance shrinking* operation that is used to filter out instances with relatively low marginal improvement to the current model; thus the computationally intensive gradient computations are performed on informative instances as much as possible. Specifically, we train a very lightweight Assistant model jointly with the original deep network, which we refer to as the Boss. The Assistant model is designed to gauge the importance of a given instance with respect to the *current* Boss model such that the shrinking operation can be applied in the batch generator. With careful design, we train the Boss and Assistant in a non-blocking and asynchronous fashion such that overhead is minimal. To demonstrate the effectiveness of AutoAssist, we conduct experiments on two contemporary applications: image classification using ResNets with varied number of layers, and neural machine translation using LSTMs, ConvS2S and Transformer models. For each application, we verify that AutoAssist leads to significant reduction in training time; in particular, $30\%$ to $40\%$ of the total operation count can be reduced which leads to faster convergence and a corresponding decrease in training time.

## 1 Introduction

Deep Neural Networks (DNNs) trained on a large number of instances have been successfully applied to many real world applications, such as [6, 11] and [20]. Due to the increasing number of training instances and the increasing complexity of deep models, variants of (mini-batch) stochastic gradient descent (SGD) are still the most widely used optimization methods because of their simplicity and flexibility. In a typical SGD implementation, a batch of instances is generated by either a randomly permuted order or a uniform sampler. Due to the complexity of deep models, the gradient calculation is usually extremely computationally intensive and requires powerful hardware (such as a GPU or TPU) to perform the entire training in a reasonable time frame. At any given time in the training process, each instance has its own utility in terms of improving the current model. As a result, performing SGD updates on a batch of instances which are sampled/generated uniformly can be suboptimal in terms of maximizing the return-on-investment (ROI) on GPU/TPU cycles. In this paper, we propose AutoAssist, a simple framework to accelerate training deep models with an Assistant that generates instances in a sequence that attempts to improve the ROI.

---

[*]The University of Texas at Austin

[†]Amazon

There have been earlier similar attempts to improve the training speed of deep learning. In [3], curriculum learning (CL), was shown to be beneficial for convergence; however, prior knowledge of the training set is required to sort the instances by its difficulty. Self-paced learning (SPL) [19] is another attempt that infers the "difficulty" of instances based on the corresponding loss value during training and decreases the training weights of these difficult instances. [13] combined the above two ideas and proposed Self Paced Curriculum learning (SPCL), which utilizes both prior knowledge and the loss values as the learning progresses. However SPCL relies on a manually chosen scheme function and introduces a considerable overhead in terms of both time and space complexity.

In our proposed AutoAssist framework, the main model, referred to as the Boss, is trained with batches generated by a light-weight Assistant which is designed to adapt to the changes in the Boss dynamically and asynchronously. Our contributions in this paper are as follows.

- We propose AutoAssist, a simple framework to accelerate training of deep neural networks by a careful designed Assistant which is able to shrink less informative instances and generate smart batches in an ROI aware sequence for the Boss to perform SGD updates.
- We also propose a concurrent computation mechanism to simultaneously utilize both CPUs and GPUs such that learning of the Boss and the Assistant are conducted asynchronously, which minimizes the overhead introduced by the Assistant.
- We conduct extensive experiments to show that AutoAssist is effective in accelerating the training of various types of DNN models including image classification using Resnets with varied number of layers, and neural machine translation using LSTMs, ConvS2S and Transformers.

## 2   Related Work

Considerable research has been conducted to optimize the way data is presented to the optimizer for deep learning. For example, curriculum learning (CL) [3], which presents easier instances to the model before hard ones, was shown to be beneficial to the overall convergence; however, prior knowledge of the training set is required to decide the curriculum. To avoid this, [28] propose to learn the curriculum with Bayesian optimization. Self-paced learning (SPL) [19] infers the difficulty of instances with the corresponding loss value and then decreases the sample weight of difficult instances. Self-paced Convolutional Networks (SPCN) [22] combines the SPL algorithm with the training of Convolutional Neural Networks to get rid of noisy data. SPL type methods generally require a user specified "pace rate" and the learning algorithm gradually incorporates more data after every epoch until the whole dataset is incorporated into the curriculum. These methods have been proven useful in a wide range of applications, including image recognition and natural language processing. Similar ideas have been developed when optimizing other machine learning models. For example, in classical SVM models, methods have been proposed to ignore trivial instances by dimension shrinking in dual coordinate descent [12].

Importance sampling is another type of method that has been proposed to accelerate SGD convergence. In importance sampling methods, instances are sampled by their importance weights. [31] proposed Iprox-SGD that uses importance sampling to achieve variance reduction. The optimal importance weight distribution to reduce the variance of the stochastic gradient is proved to be the gradient norm of the sample, see [24, 31, 1]. Despite the variance reduction effect, importance sampling methods tend to introduce large computational overhead. Before each stochastic step, the importance weights need to be updated for all instances which makes importance sampling methods infeasible for large datasets. [16] proposed an importance sampling scheme for deep learning models; however, in order to reduce computation cost for evaluating importance scores, the proposed algorithm applied a sub-sampling technique, thus leading to reliance on outdated importance scores during training. The online batch selection method [23] samples instances with the probability exponential to their last known loss value.

There are also several recent methods that propose to train an attached network with the original one. ScreenerNet [17] trains an attached neural network to learn a scalar weight for each training instance, while MentorNet [14] learns a data-driven curriculum that prevents the main network from over-fitting. Leaning to teach [9] uses a student and a teacher model and optimizes the framework with reinforcement learning. Since the additional model is another deep neural network, the above methods introduce substantial computational and memory overhead to the original training process.

# 3 A Motivating Example: SGD with Instance Shrinking for Linear SVMs

In this section, we present and analyze the theoretical and empirical properties of SGD with instance shrinking for linear Support Vector Machines (SVM). Although the analysis and observations apply to convex problems such as linear SVM, the learning from this section is the inspirational cornerstone for the design of our AutoAssist framework to accelerate training of non-convex deep learning models in Section 4.

The shrinking strategy is a key technique widely used in many popular ML software libraries to accelerate training for large-scale SVMs, such as SVM$^{Light}$ [15, Section 11.4], LIBSVM [4, Section 5.1] and LIBLINEAR [8, Section E.5]. The main idea behind the shrinking strategy is to identify variables that are *unlikely* to change in upcoming iterations and temporarily remove them from consideration to form a smaller/simpler optimization problem. Although, in most existing literature, the shrinking strategy is applied to accelerate coordinate descent based methods for the dual SVM problem, it is natural to ask if we can extend the shrinking strategy to accelerate stochastic gradient descent based methods. Due to the primal-dual relationship for SVM problems, there is a direct connection between a coordinate update in the dual SVM formulation and the stochastic gradient update with respect to the corresponding instance in the primal SVM formulation (See for example [12, Section 4.2]). After a careful examination of the relationship and the shrinking strategy adopted for dual SVMs, it can be seen that when a dual variable meets the shrinking criterion during training, the corresponding instance is not only correctly classified by the current model but is also relatively far away from the decision boundary. Furthermore, as the decision boundary changes dynamically during training, there is a mechanism for each shrunk dual variable to become active in all the aforementioned approaches, thus guaranteeing convergence. Now, we discuss a simple *Instance Shrinking* strategy designed for SGD on SVM-like convex functions.

Given a dataset $\{(\boldsymbol{x}_i, y_i) : i = 1, \ldots, N\}$, we consider a objective function parametrized as follows:

$$\min_{\boldsymbol{w} \in \mathbb{R}^d} F(\boldsymbol{w}) := \frac{1}{N} \sum_{i=1}^{N} f_i(\boldsymbol{w} \mid \boldsymbol{x}_i, y_i), \tag{1}$$

where $f_i(\cdot)$ is a loss function for the $i$-th instance. In a typical SGD method, at the $k$-th step, an instance $(\boldsymbol{x}_i, y_i)$ is uniformly sampled to perform the following update:

$$\boldsymbol{w}^{k+1} \leftarrow \boldsymbol{w}^k - \eta_k \nabla f_i(\boldsymbol{w}^k \mid \boldsymbol{x}_i, y_i), \tag{2}$$

where $\eta_k$ is the learning rate at the $k$-th step. Motivated by the above primal-dual connection for linear SVMs, in order to extend the shrinking strategy for SGD, it is intuitive to introduce the concept of *utility* of each instance for the current model $\boldsymbol{w}^k$ denoted by utility$(\boldsymbol{x}_i, y_i \mid \boldsymbol{w}^k)$, which is used to estimate the marginal improvement for this instance using the current model. As a result, we can apply the following utility-aware instance shrinking strategy to SGD:

$$\boldsymbol{w}^{k+1} \leftarrow \begin{cases} \boldsymbol{w}^k - \eta_k \nabla f_i(\boldsymbol{w}^k \mid \boldsymbol{x}_i, y_i) & \text{if utility}(\boldsymbol{x}_i, y_i \mid \boldsymbol{w}^k) \geq T_k, \\ \boldsymbol{w}^k & \text{otherwise,} \end{cases} \tag{3}$$

where $T_k$ is a threshold used to control the aggressiveness of our instance shrinking strategy.

## 3.1 Choice of the Utility Function

The effectiveness of the instance shrinking strategy for SGD depends on the choice of the utility function. There are two considerations:

- The utility function should be designed such that it accurately approximates the exact marginal improvement of the instance $(\boldsymbol{x}_i, y_i)$ for the current model, which can be defined as $F(\boldsymbol{w}^k - \eta_k \nabla f_i(\boldsymbol{w}^k \mid \boldsymbol{x}_i, y_i)) - F(\boldsymbol{w}^k)$.
- The utility function should also be simple to compute so that its overhead is minimal. As a result, the exact marginal improvement cannot be an effective utility function due to its high $O(Nd)$ computational overhead.

Obviously, the balance between both considerations is the key to designing an effective utility function. Inspired by the existing shrinking strategy used in SVM optimization, there are two simple candidates for the utility function: 1) the norm of the gradient: utility$(\boldsymbol{x}_i, y_i \mid \boldsymbol{w}^k) = \|\nabla f_i(\boldsymbol{w}^k \mid \boldsymbol{x}_i, y_i)\|$,

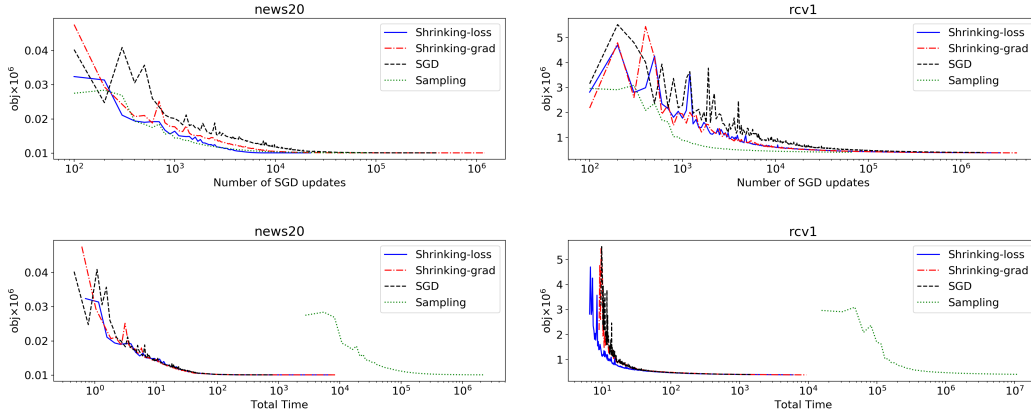

Figure 1: Comparison of Instance Shrinking with loss (*Shrinking-loss*) or gradient (*Shrinking-grad*) norm as utility and Importance Sampling for SGD on the linear SVM problem on the public *news20* and *rcv1* datasets. In terms of the number of parameter updates, instance shrinking and importance sampling strategies show faster convergence than plain SGD. Because of large computational overhead, importance sampling strategy is not effective in terms of reducing training time.

2) the loss: utility$\big(\boldsymbol{x}_i, y_i \mid \boldsymbol{w}^k\big) = f_i(\boldsymbol{w}^k \mid \boldsymbol{x}_i, y_i)$. First, both choices rely only on local information $(\boldsymbol{x}_i, y_i)$, i.e., no other instances are involved in the computation. Thus the overhead is small. Next, both choices are a good proxy: the gradient norm measures the magnitude of the change in the SGD update while the loss directly measures the performance of the current model on this instance. Experimental results on SVMs indicate that both gradient norm and loss utility achieves faster convergence. As stated below, the choice of gradient norm can be shown to be theoretically sound.

**Theorem 1.** *Given $\mu$ strongly convex function $F(\boldsymbol{w}) := \frac{1}{N} \sum_{i=1}^{N} f_i(\boldsymbol{w})$ that satisfies Property 1 in Appendix A. When $\big\| \nabla f_i(\boldsymbol{w}^k \mid \boldsymbol{x}_i, y_i) \big\|$ is used as the utility function, there exists threshold $T_k$ such that SGD with the instance shrinking update rule* (3) *converges as follows:* $\mathbb{E}\Big( \big\| \boldsymbol{w}^k - \boldsymbol{w}^* \big\|^2 \Big) \leq \frac{L}{k}$ *for some constant L, where $\boldsymbol{w}^*$ is the optimal solution of* (1).

The proof of Theorem 1 can be found in the Appendix. From Theorem 1, we can see that shrinking of instances with low utility during the training process does not hinder the theoretical convergence of SGD for strongly convex problems.

## 3.2 Practical Consideration: Computational Overhead

As mentioned earlier, computational overhead is one of the major considerations for a shrinking strategy to be effective in terms of acceleration of training process. In Figure 1, we show the results of various acceleration techniques for SGD on the linear SVM problem with two datasets: news20 and rcv1. To further demonstrate the consequence of overhead in practical effectiveness, we also include the *importance sampling* strategy into our comparison. The theoretical benefit of importance sampling for SGD has been extensively studied in the literature [24, 31, 1]. In particular, it is well known that the optimal distribution for the importance sampling strategy is that each instance be sampled with a probability proportional to the norm of gradient of this instance given the current model $\boldsymbol{w}^k$. To have a fair comparison, we implement Pegasos [27] as our plain SGD algorithm, the shrinking strategy with the loss and gradient norm as the utility function, and the importance sampling strategy with the exact optimal distribution in C++. From the top part of Figure 1, we can see that instance shrinking with both utility choice and importance sampling yields faster convergence than plain SGD in terms of the number of updates. However, in terms of the actual training time, from the bottom part of Figure 1, we can see that the importance sampling strategy is significantly slower than even plain SGD due to the huge computational overhead to maintain the exact sampling distribution that leads to the optimal theoretical convergence. On the other hand, our shrinking strategies with a very light-weight extra overhead show improvement over plain SGD in terms of training time.

It is not hard to see why the improvement in Figure 1 is almost negligible for the shrinking strategies. Due to simplicity of the linear SVM and the choice of utility function (loss in this case), the time saved by the shrinking strategy is almost the same as the overhead introduced by the computation of the utility function. Based on these observations, we see that the opportunity of a shrinking strategy to be effective in accelerating the training of complicated DNN models is in designing a utility function whose overhead is significantly lower than the computation involved in a single SGD update.

## 4 AutoAssist: Training DNNs with an Automatic Assistant

Inspired by the observations in Section 3, given that a single SGD update for a DNN model is very time-consuming, we believe that there is an opportunity for a properly designed shrinking strategy to accelerate the training for a DNN model effectively. Note that in a typical SGD training process for a DNN model, there are three major components: a batch generator which collects a batch of instances to perform a (mini-)batch stochastic gradient update; a forward pass(FP) on the DNN model to evaluate the loss values on a given batch of instances; and a backward pass(BP) on the DNN model to compute the aggregated gradient for this batch so that an SGD update can be performed. The major computation cost comes from the FP and BP phases, which usually require powerful hardware such as GPU to perform the computation efficiently. This indicates that if we can skip the FP and BP computations for instances with relatively lower utility with respect to the current model, a significant amount of computation time can be saved. Thus, in AutoAssist, we propose to design an Assistant to accelerate the training of a DNN model, which we refer to as the Boss model from now on. The Assistant is a special batch generator which implements a utility aware instance shrinking strategy.

### 4.1 A Lightweight Assistant Grows with the Boss Dynamically

To design an effective Assistant, we need to take the same two considerations of the shrinking strategy into account: on one hand, Assistant should be aware of the latest capability of the Boss model so that an accurate shrinking strategy can be used; on the other hand, Assistant should be lightweight so that the overhead is as low as possible.

Due to the fact that extracting the per-instance loss in most modern implementations of batch SGD is significantly easier than the per instance gradient, we consider using only the loss to gauge the utility of a given instance for the current Boss. However, unlike the simple linear SVM model, even the forward pass in DNN training to compute the loss is very time consuming. Thus, instead of exact loss computation, we should design a lightweight Assistant model to estimate instance utility. For most applications where DNN is applied, there exist many traditional simpler ML models which still yield reasonable performance. These "shallow counterparts" of the DNN model are good candidates to approximate the chosen utility function in

---

**Algorithm 1** Assistant: Utility Aware Batch Generator

---

1: **Input:** Dataset $\{x_i, y_i\}_{i=1}^N, \gamma_k$
2: **Output:** batch $B_k \subset \{1, \ldots, N\}$
3: **Initialize:** $B_k \leftarrow \{\}$
4: **while** $|B_k| <$ batch_size **do**
5:     $i \sim \mathtt{uniformInt}(N)$
6:     $r_1 \sim \mathtt{uniform}(0, 1)$
7:     **if** $r_1 < 1 - \gamma_k$ **then**
8:         $B_k \leftarrow B_k \cup \{i\}$
9:     **else**
10:         $r_2 \sim \mathtt{uniform}(0, 1)$
11:         **if** $r_2 < g(\psi_i \mid \phi)$ **then**
12:             $B_k \leftarrow B_k \cup \{i\}$
    **return** $B_k$

---

an efficient and accurate manner. In Section 5, we will see that even with a simple linear model our Assistant is able to reduce training time significantly for real-world applications such as image classification and neural machine translation.

In AutoAssist, we use $\phi$ to denote the parameters of the shallow Assistant model, and use $g(\cdot \mid \phi)$ to denote the approximate instance utility for the current Boss model. In particular, $g(\cdot)$ is designed to model the following probability:

$$g(\psi(x_i, y_i) \mid \phi) \approx \mathbb{P}\big[\!\big[\text{utility}(x_i, y_i \mid w^k) \geq T_k\big]\!\big], \tag{4}$$

where $\psi(x_i, y_i)$ is the feature vector used in the shallow model, utility$(x_i, y_i \mid w^k)$ is the loss for the $i$-th instance with the current Boss model $w^k$, and $T_k$ is a threshold used to determine whether the marginal utility of the $i$-th instance is large enough to include it in the mini-batch. Note that $T_k$ also changes during the entire training phase to adapt to the dynamically changing Boss model. In particular, we propose $T_k$ to be an exponential moving average of the loss of instances updated

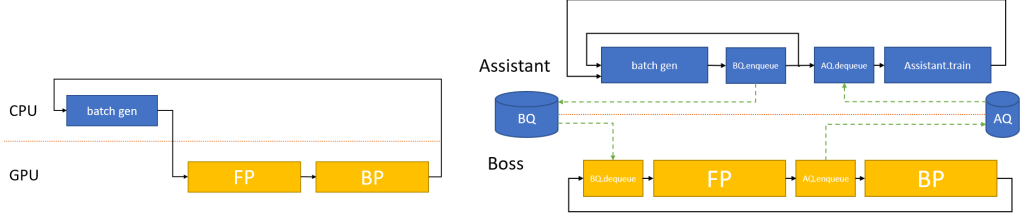

Figure 2: The sequential training scheme on the left wastes CPU/GPU cycles, while the accelerated training process with AutoAssist, shown on the right, asynchronously trains both the Boss and the Assistant leading to a more efficient use of computational resources. (AQ denotes AssistantQueue, while BQ denotes BossQueue.)

recently by Boss. There are three reasons why we choose to model the probability of the binary event instead of the instance loss directly: a) the range of the instance loss varies depending on lots of factors such as the choice of the loss function and training instance outliers. Thus, to increase the robustness of Assistant, we choose to model (4) instead; b) Shallow models usually have a limited capacity to approximate the exact loss of a given instance for a DNN model; c) with the probability output of $g(\cdot)$, Assistant can perform a "stochastic" instance shrinking strategy to avoid the situation that instances that always have a lower predicted probability are never seen by the Boss.

In order for the Assistant to know the latest capability of the Boss model, we propose a very lightweight approach to collect the latest information about the Boss. In particular, after the forward pass of a batch $B$, we collect the actual loss value (i.e., our utility function) of each instance $(\boldsymbol{x}_i, y_i)$ to form a binary classification dataset $\{(\boldsymbol{\psi}_i, z_i) : i \in B\}$, where $\boldsymbol{\psi}_i = \boldsymbol{\psi}(\boldsymbol{x}_i, y_i)$ is the feature vector, and $z_i := \mathbb{I}\big[\text{utility}\big(\boldsymbol{x}_i, y_i \mid \boldsymbol{w}^k\big) > T_k\big]$, where $\mathbb{I}[*]$ is the indicator function, is the supervision providing the latest information about the current Boss model. To keep the Assistant up-to-date with the Boss, we update the parameters for the Assistant model:

$$\boldsymbol{\phi} \leftarrow (1 - \lambda)\boldsymbol{\phi} - \eta \frac{1}{|B|} \sum_{i \in B} \nabla_{\boldsymbol{\phi}} \ell_{\text{CE}}(z_i, g(\boldsymbol{\psi}_i | \boldsymbol{\phi})), \tag{5}$$

where $\ell_{\text{CE}}$ is the cross entropy loss, $\eta$ is a fixed learning rate, and $\lambda$ is the weight decay factor.

To handle the situation where the Assistant model has not yet learned the capability of the Boss, we propose a simple mechanism to control the rate of instances, denoted by $\gamma_k \in [0, 1]$, to be passed to the stochastic instance shrinker defined by the Assistant model. In particular, in the early stage of training, we set $\gamma_0 = 0$ so that Assistant includes all instances into the batch without any shrinking operation. For the Assistant, $\gamma_k$ acts like a safeguard which takes the confidence of the current model $g(\cdot \mid \boldsymbol{\phi})$ in predicting a correct shrinking probability. The better the Assistant model performs, the higher the value $\gamma_k$ is set to. In particular, $\gamma_k$ is dynamically set to an exponential running average over the observed empirical accuracy of the Assistant model. In Algorithm 1, we describe the utility aware batch generation performed by our Assistant.

**Connections to Existing Curriculum-based Approaches.** The concept of utility of an instance in AutoAssist could be viewed as a machine learned curriculum for the current Boss. To contrast our approach, existing curriculum based approaches are not capable of evolving with the Boss model in a timely manner. For example, Self-Paced-Learning (SPL) only updates its curriculum (or self-pace) after one full epoch of the dataset. ScreenerNet [17] is another attempt to dynamically learn a curriculum with an auxiliary deep neural network, which requires additional GPU cycles to train the auxiliary DNNs causing significant overhead. We compare these two methods in Section 5.

### 4.2 An Asynchronous Computational Scheme for Joint Learning of Boss and Assistant

In traditional batch SGD training for a DNN model, as depicted in the left part of Figure 2, there is an interleaving of batch generation done in a CPU and FP/BP done in a GPU. Due to the simple logic of most existing batch generators, batch generation takes a minimal number of CPU cycles,

which causes a lengthy idle period for the CPU. The Assistant in our AutoAssist framework needs to perform instance shrinking in addition to updating the shrinking model to keep pace with Boss. To reduce the overhead, we propose an asynchronous computational scheme to fully utilize the available CPU and GPU. In particular, we maintain two concurrent queues to store the batches required for Boss and Assistant respectively:

$$\text{BossQueue} = \{\dots, B^s, \dots\} \qquad \text{and} \qquad \text{AssistantQueue} = \{\dots, M^t, \dots\}, \qquad (6)$$

where each $B^s$ is a batch of instance indices and each $M^t = \left\{(i, \text{utility}(\boldsymbol{x}_i, y_i \mid \boldsymbol{w}^k)) : i \in B^s\right\}$ is a batch of pairs of an instance index and the corresponding loss value (the utility function chosen in AutoAssist) evaluated from the forward pass of the recent Boss model on a batch $B^s$. With the help of these two concurrent queues, we can design the following computational scheme: For each GPU worker performing Boss updates, it first dequeues a batch $B^s$ from the BossQueue, performs the forward computation, enqueues the $M^t$ containing the loss values along with the instance indices to the AssistantQueue, conducts the backward computation to perform the SGD update on the parameters of the Boss model. On the other hand, for each CPU worker performing Assistant updates, whenever one of the queues is empty, it always generates and enqueues a new batch to the BossQueue; otherwise, it dequeues an $M^t$ from the AssistantQueue, forms the binary dataset from $M^t$ to perform the update (5). An illustration of the scheme is shown in the right part of Figure 2. It is not hard to see that both the CPU and GPU are utilized in our scheme, and the only overhead introduced is the step to collect the loss values and enqueuing the corresponding $M^t$, which is an operation that has minimal computational cost.

# 5  Experimental results

To demonstrate the effectiveness of AutoAssist in training large-scale DNNs, we conduct experiments [3] on two applications where DNNs have been successful: image classification and neural machine translation.

## 5.1  Image classification

**Datasets and DNN models.** We consider MNIST [21], rotated MNIST[4], CIFAR10 [18] and raw ImageNet [7] datasets. The dataset statistics are presented in Table 1 of the Appendix. For the DNN models, we consider the popular ResNets with varied number of layers (18, 34, and 101 layers).

**Experimental Setting.** Following [11], we use SGD with momentum as the optimizer. The detailed parameter settings are listed in the Appendix. Three acceleration methods and SGD baseline are included in our comparison:

- AutoAssist: L2-regularized logistic regression as our Assistant model, where the stacked pixel values of the raw image are used as the feature vector except for ImageNet, where low resolution images are used as feature vector.
- SGD baseline: vanilla stochastic gradient descent with momentum.
- Self-Paced Learning (SPL): We implement the same self-paced scheme in [22].
- ScreenerNet: We use the same screener structure and settings described in [17].

Experimental results, that include the training loss versus training time for each model and dataset combination, are shown in Figure 3. As a sanity check, we also verified that the AutoAssist reaches the same accuracy as SGD baseline. It can be observed that AutoAssist outperforms other competing approaches in all (model, dataset) combinations in terms of training time. The Assistant model is able to reach $80\% \sim 90\%$ shrinking accuracy even with a simple linear logistic regression model, as shown in Figure 6 in the Appendix. It can be seen that AutoAssist yields effective SGD acceleration compared to other approaches.

## 5.2  Neural Machine Translation

**Datasets and DNN models.** We consider the widely used WMT14 English to German dataset, which contains 4M sentence pairs. We constructed source and target vocabulary with size 42k and 40k. In

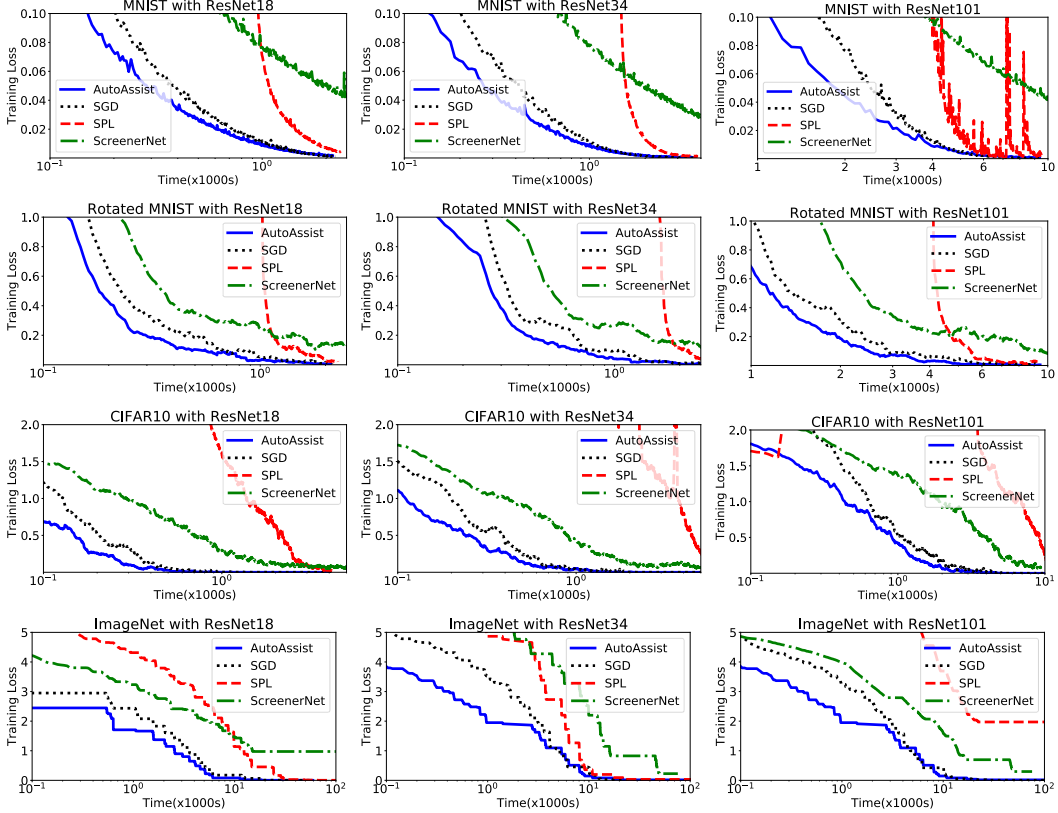

Figure 3: Comparison of various training schemes on image classification. X-axis is the training time in seconds, while Y-axis is the training loss. ResNets with varied number of layers ranging from {18, 34, 101} are considered. Each column of figures shows the ResNet results with a specific number of layers, while each row of figures shows results on a specific dataset.

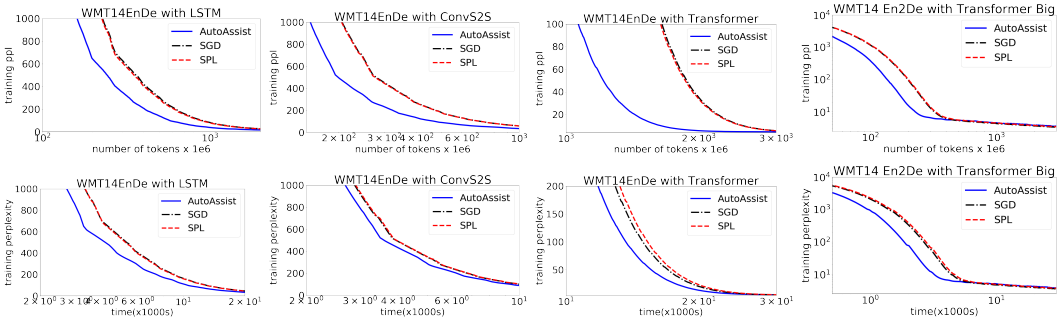

Figure 4: Comparison of various training schemes on neural machine translation. X-axis is the training time, while Y-axis is the training perplexity. Four commonly sequence-to-sequence models are considered: LSTM, ConvS2S, and Transformer(base and big). We use 8 CPUs for Assistant and 8 GPUs for Boss.

terms of the DNN models for NMT, we consider four popular deep sequence models: LSTM [30], ConvS2S [10], Transformer base and big model [29].

**Experimental Setting.** We implement AutoAssist with the asynchronous update mechanism described in Section 4.2 under the Fairseq [25] codebase. In particular we enable multiple CPUs for multiple Assistant updates and multiple GPUs for Boss training. We use 8 Nvidia V100 GPUs for

Boss training and stop after training on 6 billion tokens. As ScreenerNet described in [17] cannot be trivially extended to the NMT task, we exclude ScreenerNet in our comparison.

- AutoAssist: L2-regularized logistic regression is used as our Assistant model, where the term-frequency / inverse-document-frequency [26] of each source-target pair is used as the feature vector. The TF/IDF features are computed during preprocessing time to reduce overhead. Note that although the Boss model is trained in a data-parallel fashion (gradients are synchronized after each back-propagation), Assistant is updated in an asynchronous manner as described in Section 4.2. In order to generate batches to train the Boss model with $p$ GPUs, $p$ Assistant models are created such that batch generation can be done asynchronously for each GPU worker.
- SGD baseline: vanilla stochastic gradient descent with momentum.
- Self-Paced Learning (SPL): we implement the same self-paced scheme in [22].

The experimental results are shown in Figure 4, which includes the number of tokens/training time versus training perplexity for each DNN model. Similar to image classification, we also observe that AutoAssist outperforms other training schemes for all our experiments on neural machine translation. In particular, the AutoAssist is able to save around $40\%$ tokens per epoch and achieves better final BLEU scores than the baseline. Some training statistics and the final BLEU scores are listed in table 2 in Appendix.

## 6   Conclusions

In this paper, we propose a training framework to accelerate deep learning model training. The proposed AutoAssist framework jointly trains a batch generator (Assistant) along with the main deep learning model (Boss). The Assistant model conducts instance shrinking to get rid of trivial instances during training and can automatically adjust the criteria based on the ability of the Boss. We further propose a method to reduce the computational overhead by training Assistant and Boss asynchronously on CPU/GPU and extend this framework to multi-GPU/CPU settings. Experimental results demonstrate that both convergence speed and training time are improved by our Assistant model.

**Acknowledgement** This research was supported by NSF grants IIS-1546452 CCF-1564000 and AWS Cloud Credits for Research program.

## Footnotes

[3]The code is available at `https://github.com/zhangjiong724/autoassist-exp`

[4]Constructed by randomly rotating each image in the original MNIST within $\pm 90$ degrees.

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
