[Supplementary Material · AutoAssist-appendix-camera-ready.pdf]

# Appendices

## A Proof of Theorem 1

Consider the similar strongly convex conditions used in Pegasos [27], where $F(\boldsymbol{w})$ satisfies the following conditions:

**Property 1.** $F(\boldsymbol{w}) := \frac{1}{N} \sum_{i=1}^{N} f_i(\boldsymbol{w})$ *satisfies:*
- *$F$ is $\mu$-strongly convex,*
- $\arg \min_{\boldsymbol{w}} F(\boldsymbol{w}) \in B_D = \{\boldsymbol{w} \mid \|\boldsymbol{w}\| \leq D\}$, *and*
- $\|\nabla f_i(\boldsymbol{w})\| \leq G, \ \forall \boldsymbol{w} \in B_D$

With the step size as $\eta_k = \frac{1}{\mu k}$, shrinking threshold $T_k = \frac{G}{k}$ and initialization $w^1 \in B_D$, we can prove Theorem 1 with above assumption.

[Proof of Theorem 1] From strong convexity, we have:

$$F(\boldsymbol{w}^*) - F(\boldsymbol{w}^k) \geq \langle \nabla F(\boldsymbol{w}^k), \boldsymbol{w}^* - \boldsymbol{w}^k \rangle + \frac{\mu}{2}\|\boldsymbol{w}^k - \boldsymbol{w}^*\|^2$$

$$F(\boldsymbol{w}^k) - F(\boldsymbol{w}^*) \geq \langle \nabla F(\boldsymbol{w}^*), \boldsymbol{w}^k - \boldsymbol{w}^* \rangle + \frac{\mu}{2}\|\boldsymbol{w}^k - \boldsymbol{w}^*\|^2.$$

Adding the above inequalities gives:

$$\langle \nabla F(\boldsymbol{w}^k) - \nabla F(\boldsymbol{w}^*), \boldsymbol{w}^k - \boldsymbol{w}^* \rangle \geq \mu\|\boldsymbol{w}^k - \boldsymbol{w}^*\|^2$$
$$\Rightarrow \quad \langle \nabla F(\boldsymbol{w}^k), \boldsymbol{w}^k - \boldsymbol{w}^* \rangle \geq \mu\|\boldsymbol{w}^k - \boldsymbol{w}^*\|^2 \qquad (\text{since } \nabla F(\boldsymbol{w}^*) = 0)$$
$$\Rightarrow \quad \langle \mathbb{E}(\nabla f_i(\boldsymbol{w}^k)), \boldsymbol{w}^k - \boldsymbol{w}^* \rangle \geq \mu\|\boldsymbol{w}^k - \boldsymbol{w}^*\|^2 \tag{7}$$

Where $i$ is the data index sampled at step $k$, the update step is defined as:

$$\boldsymbol{w}^{k+1} = \boldsymbol{w}^k - \eta_k \boldsymbol{g}_i^k \tag{8}$$

where

$$\boldsymbol{g}_i^k = \nabla f_i(\boldsymbol{w}^k) - \boldsymbol{\epsilon}_i^k \tag{9}$$

$$\boldsymbol{\epsilon}_i^k = \nabla f_i(\boldsymbol{w}^k)\mathbf{I}\left(\|\nabla f_i(\boldsymbol{w}^k)\| \leq \frac{G}{k}\right) \tag{10}$$

Next, we have:

$$\mathbb{E}(\|\boldsymbol{w}^{k+1} - \boldsymbol{w}^*\|^2) = \mathbb{E}(\|\boldsymbol{w}^k - \eta_k \boldsymbol{g}_i^k - \boldsymbol{w}^*\|^2)$$
$$= \mathbb{E}(\|\boldsymbol{w}^k - \boldsymbol{w}^*\|^2) - 2\eta_k \mathbb{E}\langle \boldsymbol{g}_i^k, \boldsymbol{w}^k - \boldsymbol{w}^* \rangle + \eta_k^2 \mathbb{E}(\|\boldsymbol{g}_i^k\|^2)$$
$$\leq \mathbb{E}(\|\boldsymbol{w}^k - \boldsymbol{w}^*\|^2) - 2\eta_k \mathbb{E}\langle \nabla f_i(\boldsymbol{w}^k), \boldsymbol{w}^k - \boldsymbol{w}^* \rangle$$
$$+ \eta_k^2 G^2 + 2\eta_k \mathbb{E}\langle \boldsymbol{\epsilon}_i^k, \boldsymbol{w}^k - \boldsymbol{w}^* \rangle \tag{11}$$

The last term from (11) can be upper bounded as follows:

$$2\eta_k \mathbb{E}\langle \boldsymbol{\epsilon}_i^k, \boldsymbol{w}^k - \boldsymbol{w}^* \rangle$$
$$= 2\eta_k \mathbb{E}\left(\langle \nabla f_i(\boldsymbol{w}^k), \boldsymbol{w}^k - \boldsymbol{w}^* \rangle \mathbf{I}\left(\|\nabla f_i(\boldsymbol{w}^k)\| \leq \frac{G}{k}\right)\right)$$
$$\leq 2\eta_k \mathbb{E}\left(\|\nabla f_i(\boldsymbol{w}^k)\|\|\boldsymbol{w}^k - \boldsymbol{w}^*\|\mathbf{I}\left(\|\nabla f_i(\boldsymbol{w}^k)\| \leq \frac{G}{k}\right)\right)$$
$$\leq 2\eta_k \frac{G}{k}\mathbb{E}(\|\boldsymbol{w}^k - \boldsymbol{w}^*\|) \tag{12}$$

Substituting (7) and (12) into (11), we have:

$$\mathbb{E}(\|\boldsymbol{w}^{k+1} - \boldsymbol{w}^*\|^2) \leq \mathbb{E}(\|\boldsymbol{w}^k - \boldsymbol{w}^*\|^2)\left(1 - \frac{2}{k}\right) + \frac{G^2}{\mu^2 k^2} + \frac{2G}{\mu k^2}\mathbb{E}(\|\boldsymbol{w}^k - \boldsymbol{w}^*\|) \tag{13}$$

Letting $\hat{D} = \max\left(D, \frac{2G}{\mu}\right)$ and letting $L = 4\hat{D}^2$, we have:

$$L = 4\hat{D}^2 \geq \frac{G^2}{\mu^2} + \frac{4\hat{D}G}{\mu} \tag{14}$$

The convergence can be established by induction. We want to proof the following condition holds for $k \geq 1$:

$$\mathbb{E}(\|\boldsymbol{w}^k - \boldsymbol{w}^*\|^2) \leq \frac{L}{k} \tag{15}$$

When $k = 1$, the inequality hold:

$$\mathbb{E}(\|\boldsymbol{w}^1 - \boldsymbol{w}^*\|^2) \leq \frac{L}{1} \tag{16}$$

Suppose the same holds for $k$. Then for $k + 1$, using (13) we have:

$$\mathbb{E}(\|\boldsymbol{w}^{k+1} - \boldsymbol{w}^*\|^2) \leq \left(1 - \frac{2}{k}\right)\mathbb{E}(\|\boldsymbol{w}^k - \boldsymbol{w}^*\|^2) + \frac{G^2}{\mu^2 k^2} + \frac{2G}{\mu k^2}\mathbb{E}(\|\boldsymbol{w}^k - \boldsymbol{w}^*\|) \tag{17}$$

Since $\mathbb{E}(\|\boldsymbol{w}^k - \boldsymbol{w}^*\|) \leq \sqrt{\mathbb{E}(\|\boldsymbol{w}^k - \boldsymbol{w}^*\|^2)} \leq \sqrt{L} \leq 2\hat{D}$, we have:

$$\mathbb{E}(\|\boldsymbol{w}^{k+1} - \boldsymbol{w}^*\|^2) \leq \left(1 - \frac{2}{k}\right)\frac{L}{k} + L\frac{1}{k^2} \tag{18}$$

$$= \frac{k-1}{k^2}L \leq \frac{L}{k+1} \tag{19}$$

Thus according to the principle of mathematical induction, (15) hold for all $k \geq 1$ ∎

In many machine learning models, the calculation of the objective function value takes many fewer operations than the computation of norm of the gradient. Also, for $\mu$-strongly convex $F(\boldsymbol{w})$ which is $M$-Lipschitz smooth we have:

$$\mu(F(\boldsymbol{w}) - F(\boldsymbol{w}^*)) \leq \frac{1}{2}\|\nabla F(\boldsymbol{w})\|^2 \leq \frac{M^2}{\mu}(F(\boldsymbol{w}) - F(\boldsymbol{w}^*)),$$

where the left inequality follows from the Polyak-Lojasiewicz inequality, showing that the gradient norm can be bounded by the loss function value. Due to these reasons, in practice we can also use the loss function value as the utility of shrinking.

## B   Algorithms for CPU/GPU parallelization

---
**Algorithm 2** Assistant (CPU)

---
- **Input:** Training dataset $D = \{\boldsymbol{x}_i, \boldsymbol{y}_i\}_{i=1}^N$, BossQueue min size $c$
- **Initialize:** BossQueue, AssistantQueue, Assistant
- While **True**:
    - If BossQueue.size()$< c$:
        * B = Assistant.sample_batch()
        * BossQueue.enqueue(B)
    - Else if not AssistantQueue.empty():
        * **M**= AssistantQueue.pop()
        * grad = Assistant.gradient(**M**)
        * Assistant.update(grad)

---

## C   Discussion on Computational Overhead of Importance Sampling

The case of importance sampling in Figure 1 gives an example that too much overhead could ruin the acceleration effect. The importance sampling algorithm uses a precise estimation of instance utility (normalized gradient norm) which contains both local and global information. However, as the model changes after every parameter update, importance weights need to be updated, which

**Algorithm 3** Boss (GPU)

- **Input:**
- **Initialize:** Boss
- While **True**:
  - If not BossQueue.empty():
    * B = BossQueue.pop()
    * M = Boss.Forward(B)
    * AssistantQueue.enqueu(M)
    * grad = Boss.Backward(M)
    * Boss.update(grad)

Table 1: Image Classification datasets statistics

| Dataset | # instances | image size | # classes |
|---|---|---|---|
| MNIST | 60,000 | $28 \times 28$ | 10 |
| Rotated MNIIST | 60000 | $28 \times 28$ | 10 |
| CIFAR10 | 60,000 | $32 \times 32 \times 3$ | 10 |
| ImageNet | 1,281,167 | $224 \times 224 \times 3$ | 1,000 |

introduces large computational overhead. A pre-sampling technique is sometimes used to tackle this issue. First a subset of data $C \subset [N]$ (with $|C| \ll N$) is uniformly sampled and importance scores are evaluated only on $C$. After training enough number of batches on $C$, another chunk is sampled and evaluated. This can reduce the computational overhead but may introduce new issues. Firstly, the importance weights are fixed once evaluated and may be outdated after parameter updates. In real applications with large data, the model can evolve substantially even within one epoch through the data. Secondly, substantial computational overhead is introduced even with the sub-sampling technique. This indicates that any acceleration method will fail on SGD if it requires acces to global information.

# D   Experimental parameter setting and further results

**Experimental settings for image classification.** For the image classification experiments reported in Figure 3, we set the learning rate for MNIST, rotated MNIST and raw ImageNet to be 0.0005, 0.005 for CIFAR10. The batch size is chosen to be 16 for MNIST/rotated MNIST and CIFAR10 datasets and 64 for raw ImageNet. As raw ImageNet dataset cannot be load into memory at once, to reduce the image preprocessing cost, we preprocessed all the images and pre-stored them into 12 data chunks. During training, the model loads and trains on one chunk at a time. For SPL algorithm, we implemented the same self-paced scheme in [22] with $q(t) = 4$ and $maxgen = 20$.

**Experimental setting for NMT task.** We train WMT14 dataset on 8 Tesla V100 GPUs with around 40k tokens per training batch (5k tokens per batch per GPU). Each model is trained until 6 billion tokens are seen. The BLEU scores are evaluated with the averaged model of last 10 checkpoints (epochs). For the Transformer big model, to allocate memory for 3000 tokens per GPU, we use half precision (16 bit) floating point type.

Table 2: Statistics of the Transformers on WMT14 dataset. The model used for computing BLEU score is the averaged model of last 10 epochs.

| model | # tokens per epoch | time per epoch(s) | final BLEU score |
|---|---|---|---|
| **SGD(base)** | 112.1M | 1115 | 27.25 |
| **SPL(base)** | 112.1M | 1142 | 27.32 |
| **AutoAssist (base)** | 72.6M | 973 | **27.42** |
| **SGD(big)** | 112.1M | 1190 | 27.41 |
| **SPL(big)** | 112.1M | 1221 | 27.40 |
| **AutoAssist (big)** | 70.8M | 1016 | **27.63** |

Figure 5: Comparison of various SGD acceleration approaches on image classification. $x$-axis is the training time in seconds, while $y$ axis is the test accuracy. ResNets with varied depth ranging from $\{18, 34, 101\}$ are considered. Each block column of figures show the results of the ResNet with a specific number of layers. Each block row of images show the results on a specific dataset.

Figure 6: The Assistant predictive accuracy and safeguarding rate $(1 - \gamma)$. Even with the simplest logistic regression model, the Assistant is able to reach up to $90\%$ accuracy while predicting the shrinking label, resulting in a confident Assistant with low safe-guarding rate $(1 - \gamma)$.