[Reviews · NeurIPS 2019]

Reviewer 1



The theoretical study of instance shrinkage in pegasos is as far as I know novel and interesting. Specially interesting is how instance shrinkage does not affect the solution the model converges to, which justifies later experiments which ignore importance sampling in deep nets. Similarly, the idea of training a small assistant model just to predict the loss of the base model on unseen examples is straightforward and potentially useful. The algorithm is clearly described, including all hyperparameters, and it does look like it should be possible to replicate the experiments. It's unclear from reading the experimental section, however, that this algorithm is actually an improvement over just regular training with no curriculum attached. No experiment in the paper with a deep net shows a learning curve for training with no curriculum, so it's unclear whether adding an assistant actually helps at all, or by how much. The paper is also fairly light on how the hyperparameters were tuned; ideally we'd see that the hyperparameters were tuned on baseline SGD/momentum to minimize convergence time to a given target accuracy; then we'd plot the performance of the autoassist method next to the performance of the tuned SGD/momentum baseline and other curriculum learning methods. The paper also completely omits any analysis on how different characteristics of the assistant model affect the performance of autoassist. Some questions I'd like to understand better are: 1. How simple can the assistant model be to still observe an improvement in training time? 2. How aggressive can the example filtering be until convergence is affected? 3. What is the ideal complexity of the assistant model? Does it make sense to have an assistant almost as complex as the base model itself? 4. Is the behavior of auto assist affected by the batch size used? 5. Some learning tasks (like cifar) are often overfit by state-of-the-art models, while others (like lm1b language modeling) are underfit; does autoassist behave differently in the overfit vs underfit regime? (specifically, does autoassist degrade in performance as the training loss goes to zero on all training set examples, as happens on large nets on cifar?) As the paper currently stands it's hard to judge whether autoassist is an improvement even over baseline SGD/momentum, and it's impossible to tell how autoassist would behave in practice. I don't think all questions above need to be answered to make this paper acceptable, but some information around more of them would be very helpful. ---- after reading author feedback I revise my score to 7, assuming the authors clarify the paper to cover the points I raised.

Reviewer 2



Much of my comments are reflected in section 1 (Contributions) and section 5 (Improvements). Just provide more points here: 1. In general, the problem this paper wishes to address is super important, especially when we are in the era of big model + big data (such as BERT). The point out from which this paper tries to solve this problem is also making sense: try to abandon the useless data points and only focus on the important one. In particular, I like the idea of using shallow model as a proxy to the true "utility" and cast the training process as a binary classification task. 2. The paper is also presented in a reasonably clear manner. 3. I have major concerns towards the experimental setup, most of which could be reflected in section 5 of this review. Apart from these points, the improvements seem not that significant, especially on the WMT14 dataset (a larger scale one) in Fig.4. And I would suggest that moving Fig. 6 in appendix to the main text since we care about the final evaluation measure (e.g., accuracy and BLEU) much more than training loss/ppl. 4 Several representative related works are ignored in the paper. Indeed there is a rich literature that talks about using importance sampling to boost ML model training that is largely missing. For example, please check paper [1-4]. [1] Loshchilov, I. and Hutter, F., 2015. Online batch selection for faster training of neural networks. arXiv preprint arXiv:1511.06343. [2] Fan, Y., Tian, F., Qin, T., Li, X.Y. and Liu, T.Y., 2018. Learning to Teach. ICLR [3] Tsvetkov, Y., Faruqui, M., Ling, W., MacWhinney, B. and Dyer, C., 2016, August. Learning the Curriculum with Bayesian Optimization for Task-Specific Word Representation Learning. In Proceedings of the 54th Annual Meeting of the Association for Computational Linguistics (Volume 1: Long Papers) (pp. 130-139). [4] I also remember a paper appearing in NIPS 2017 about dynamic and automatic batch selection towards NN, but sorry for that I do not remember the exact name. Will update the review once I found that paper. ________________________________________________________________________________________________________ Post rebuttal: I thank the author's response. I still do not think that a paper claiming faster training convergence could establish itself in NeurIPS, if 1) no convincing results on large scale tasks are demonstrated; 2) no thorough discussion/comparison with the literature is provided.

Reviewer 3



The paper proposes a lightweight assistant model to filter training samples. The lightweight model is learned online with the "Boss" model. The paper is well written and technically sound. Experiments show higher speedup over previous works. *** update Keep the original rate and please "We will include raw ImageNet in the revised version."

[Author Response · NeurIPS 2019]

We thank the reviewers for their comments. We will carefully modify the paper according to the suggestions.

Figure 1: Comparison of different learning schemes on RotMNIST classification and IWSLT translation tasks.

**Comparison with the vanilla SGD baseline.**    In the current manuscript, we only focused on the comparison on
curriculum-based approaches (SPL [18] and ScreenerNet [16]) which demonstrates superiority over plain SGD and
uses the final accuracy of the SGD as a sanity check for the quality of models trained with AutoAssist (e.g.g, BLEU
score results in Appendix D). We thank and agree with the reviewers for the constructive suggestion that demonstration
of superior performance of AutoAssist over the vanilla SGD without a curriculum would make the paper stronger. We
will add this SGD baseline to the all experiments in the revised version. Due to the limited time of the rebuttal period,
only partial results are included in Figure 1. Among the three curriculum-based models and the non-curriculum SGD
baseline, we can see that AutoAssist shows the best performance.

**To reviewer 1:**
– We experimented with learning rates from $1^{-4}$ to $5^{-3}$ and pick the one with the best performance for all three models.
For the NMT tasks, we used the same parameter settings from previous papers, as described in section 5.2. We've done
sanity check with baseline SGD that the setting can reach the similar BLEU score as reported in the original paper.

– The linear assistant model we used is already one of the simplest ML models for the given format, and we still observe
better performance.

– The aggressiveness is decided by Assistant through the safeguard variable $\gamma$, which is discussed in section 4.1 and
plotted in Figure 2(R).

– It is true that a more complex Assistant model might yield a better shrinking accuracy. However, in order to minimize
the time overhead, it is better to choose an Assistant model such that it requires less batch training time on CPU than
the training time of Boss on GPU so that Assistant training can be hidden behind GPU training.

– We've experimented with batch size varying from 16 to 128 for image classification tasks and number of tokens
varying from 1000 to 5000 for NMT tasks. Assistant model shows similar performance over different batch sizes.

– As described in previous paragraph, even with zero loss there is still a safeguarding base possibility that the instance
will be incorporated into a training batch. In the extreme case that all training loss go to zero, the Assistant will gradually
reduce to uniform sampling.

**To reviewer 2:**
– The reason why we didn't use raw ImageNet is that there is no available implementation for ScreenerNet on ImageNet
dataset. However, we will provide results on raw ImageNet dataset and large Transformer model in the revised version.

– More ablation study will be included in our revision. The comparison of the portion of batch generation time over the
entire training time is showed in Figure 2(L). With the parallel framework, the batch generation can be done in parallel
to the GPU training and thus result in similar percentage of time used for batch generation ($3\%$ to $4\%$) in the plain SGD.

– Since the Assistant is learning from an evolving Boss model, it will not converge to a stationary point as most linear
models do. However, we do observe the Assistant usually reaches a relatively stable state with high accuracy ($\sim 90\%$),
during the first several epochs of the Boss (please see the blue carve in Figure 2(R)).

– We will include the listed related works into discussion.

**To reviewer 3:**
– In most cases while the data can be preprocessed
and loaded into the memory, the CPU has idle periods.
However, it is true that in the case that CPU needs
to load every training batch from disk (such as raw
ImageNet), the CPU utilization could be high. We will
make this more clear in the revised version.

– As long as the CPU training is faster than the GPU
training, the overhead could be hide behind GPU time,
thus a shallow CNN is plausible.

– We will include raw ImageNet in the revised version.

Figure 2: (L) Percentage of time used for batch generation during training. (R) Assistant predictive accuracy and safeguarding rate $(1 - \gamma)$.

[Meta-Review · NeurIPS 2019]

This paper addresses an important problem and the empirical results look promising. The method is simple and clearly presented. For making this work more convincing, as pointed out by the reviewers, it would be nice to add tuned SGD/momentum baseline, and have a thorough discussion with related work.